# The Melanocortin Receptor Accessory Protein 2 promotes food intake through inhibition of the Prokineticin Receptor-1

**Anna L Chaly[1,3,2†], Dollada Srisai[1,3,2†], Ellen E Gardner[1,3,2], Julien A Sebag[1,3,2*]**

[1]Department of Molecular Physiology and Biophysics, Carver College of Medicine, University of Iowa, Iowa City, United States; [2]Pappajohn Biomedical Institute, University of Iowa, Iowa City, United States; [3]Fraternal Order of Eagle Diabetes Research Center, University of Iowa, Iowa City, United States

**Abstract** The Melanocortin Receptor Accessory Protein 2 (MRAP2) is an important regulator of energy homeostasis and its loss causes severe obesity in rodents. MRAP2 mediates its action in part through the potentiation of the MC4R, however, it is clear that MRAP2 is expressed in tissues that do not express MC4R, and that the deletion of MRAP2 does not recapitulate the phenotype of *Mc4r* KO mice. Consequently, we hypothesized that other GPCRs involved in the control of energy homeostasis are likely to be regulated by MRAP2. In this study we identified PKR1 as the first non-melanocortin GPCR to be regulated by MRAP2. We show that MRAP2 significantly and specifically inhibits PKR1 signaling. We also demonstrate that PKR1 and MRAP2 co-localize in neurons and that *Mrap2* KO mice are hypersensitive to PKR1 stimulation. This study not only identifies new partners of MRAP2 but also a new pathway through which MRAP2 regulates energy homeostasis.

*For correspondence: julien-sebag@uiowa.edu

†These authors contributed equally to this work

**Competing interests:** The authors declare that no competing interests exist.

## Introduction

G-Protein Coupled Receptors (GPCRs) are responsible for a wide variety of physiological functions, and their activity is precisely regulated, not only by their ligands (agonists and in some cases antagonists) but also by other factors including G-proteins, β-arrestins, kinases, lipids, ions and accessory proteins (*van der Westhuizen et al., 2015*). GPCR accessory proteins are single-pass transmembrane proteins that regulate trafficking and/or signaling of the receptors to which they bind (*Cooray et al., 2009*). A few examples include: the receptor activity modifying proteins (RAMPs), which promote the trafficking and ligand specificity of the calcitonin receptor-like receptor/adrenomedullin receptor and several other GPCRs (*Parameswaran and Spielman, 2006*; *Sexton et al., 2006*); the receptor transporting proteins (RTPs), which promote the trafficking of odorant and taste receptors (*Saito et al., 2004*); and the melanocortin receptor accessory proteins (MRAPs), whose known roles prior to this study were limited to the regulation of trafficking and signaling of melanocortin receptors (*Metherell et al., 2005*; *Sebag and Hinkle, 2007*; *2009a*; *Sebag et al., 2013*; *Cerdá-Reverter et al., 2013*; *Asai et al., 2013*).

Two MRAPs exist in mammals. MRAP1 is expressed in few tissues like adipocytes (*Xu et al., 2002*) and the adrenal glands, where it is essential for proper trafficking and signaling of the melanocortin-2 receptor (*Metherell et al., 2005*; *Sebag and Hinkle, 2007*; *Sebag and Hinkle, 2009a*; *2009b*). MRAP2 is likewise expressed in the adrenal glands, but also in the brain and other tissues. Unlike MRAP1, the physiological functions and binding partners of MRAP2 are largely unknown. MRAPs display a remarkable topology: they are inserted in the plasma membrane in both the $N_{out}/C_{in}$ and $N_{in}/C_{out}$ orientation, with MRAPs of opposite orientation forming anti-parallel homodimers

**eLife digest** The brain plays a major role in controlling how much food animals eat. The nerve cells (neurons) involved in this process contain "receptors" that respond to cues from various parts of the body. For example, a receptor called PKR1 acts to limit food intake. The activities of PKR1 and other receptors are tightly regulated in cells, but it is not clear how this works.

A protein called MRAP2 is known to regulate the activity of a receptor that regulates food intake and energy use in the brain. However, MRAP2 may also interact with other receptors to control food intake. Here, Chaly, Srisai et al. investigated whether MRAP2 can regulate the activity of PKR1 in animal cells and rodents.

The experiments show that MRAP2 can interact with and inhibit the activity of PKR1. Furthermore, both MRAP2 and PKR1 can be found in the same neurons. Mutant mice that lack the gene that encodes MRAP2 have higher levels of PKR1 activity and eat less than normal mice when PKR1 is stimulated. Together the experiments suggest that MRAP2 can increase food intake by preventing PKR1 from being activated in the brain. The next steps are to find out if this protein regulates other receptors involved in the control of food intake, and to test whether PKR1 and MRAP2 also play a role in regulating energy usage.

(*Sebag and Hinkle, 2007*; *2009b*). This structure is important for MRAP function (*Sebag and Hinkle, 2009b*), and so far appears to be unique to these proteins in the eukaryotic proteome.

MRAP2 is an important regulator of energy homeostasis and *Mrap2* KO mice develop severe obesity (*Asai et al., 2013*). The mechanisms through which MRAP2 regulates energy balance have not yet been fully identified, however, they include the potentiation of the melanocortin-4 receptor (MC4R) (*Sebag et al., 2013*; *Asai et al., 2013*), a protein central to the regulation of food intake and energy expenditure. Notably, like their *Mrap2* KO counterparts, *Mc4r* KO mice are severely obese (*Butler and Cone, 2003*). There are however key differences between the obesity phenotypes of the two strains. In particular, the *Mc4r* KO mice are hyperphagic, have decreased energy expenditure and are insulin resistant (*Butler and Cone, 2002*; *2003*), characteristics that are absent in the *Mrap2* KO mice (*Asai et al., 2013*). These phenotypic differences suggest that MC4R is not the only effector through which MRAP2 regulates the energy state, a conclusion consistent with the fact that MRAP2 is expressed in tissues that do not express MC4R (*Asai et al., 2013*).

Food intake is regulated by the activity of several GPCRs including the prokineticin receptor 1 (PKR1). Activation of PKR1 in vivo, through central or peripheral injection of its ligand prokineticin 2 (PK2), was shown to significantly decrease food intake (*Gardiner et al., 2010*; *Beale et al., 2013*). In addition to food intake, PKR1 plays important roles in the regulation of a variety of physiological functions including energy expenditure (*Zhou et al., 2012*), insulin sensitivity (*Dormishian et al., 2013*), gastrointestinal contraction (*Li et al., 2001*), nociception (*Negri and Lattanzi, 2011*), cardiovascular function and angiogenesis (*Boulberdaa et al., 2011*; *Urayama et al., 2007*). Meanwhile, its orthologue PKR2 regulates placentation (*Hoffmann et al., 2007*), inflammation (*Denison et al., 2008*) and nociception (*Negri and Lattanzi, 2011*). PKR1 and 2 couple to both the Gαs and Gαq proteins (*Ngan and Tam, 2008*), and consequently signal through the cAMP as well as the IP3/calcium pathways. Even though PKR1 and PKR2 appear to have some redundant physiological functions, it was shown that only PKR1 regulates food intake since injection of PK2 retains its full anorexigenic effect in PKR2 KO mice but does not decrease food intake in PKR1 KO mice (*Beale et al., 2013*).

In this study we identify PKR1 as the first non-melanocortin receptor to be regulated by MRAP2 and discover a novel mechanism of regulation of energy homeostasis by MRAP2 through the modulation of PKR1 signaling.

## Results

For PKR1 signaling to be regulated by MRAP2 in-vivo, the latter needs to be expressed along with the receptor. To determine what organs express both proteins, we performed RT-PCR on mRNA extracted from several mouse tissues. MRAP2 was readily detectable in the brain (hypothalamus and

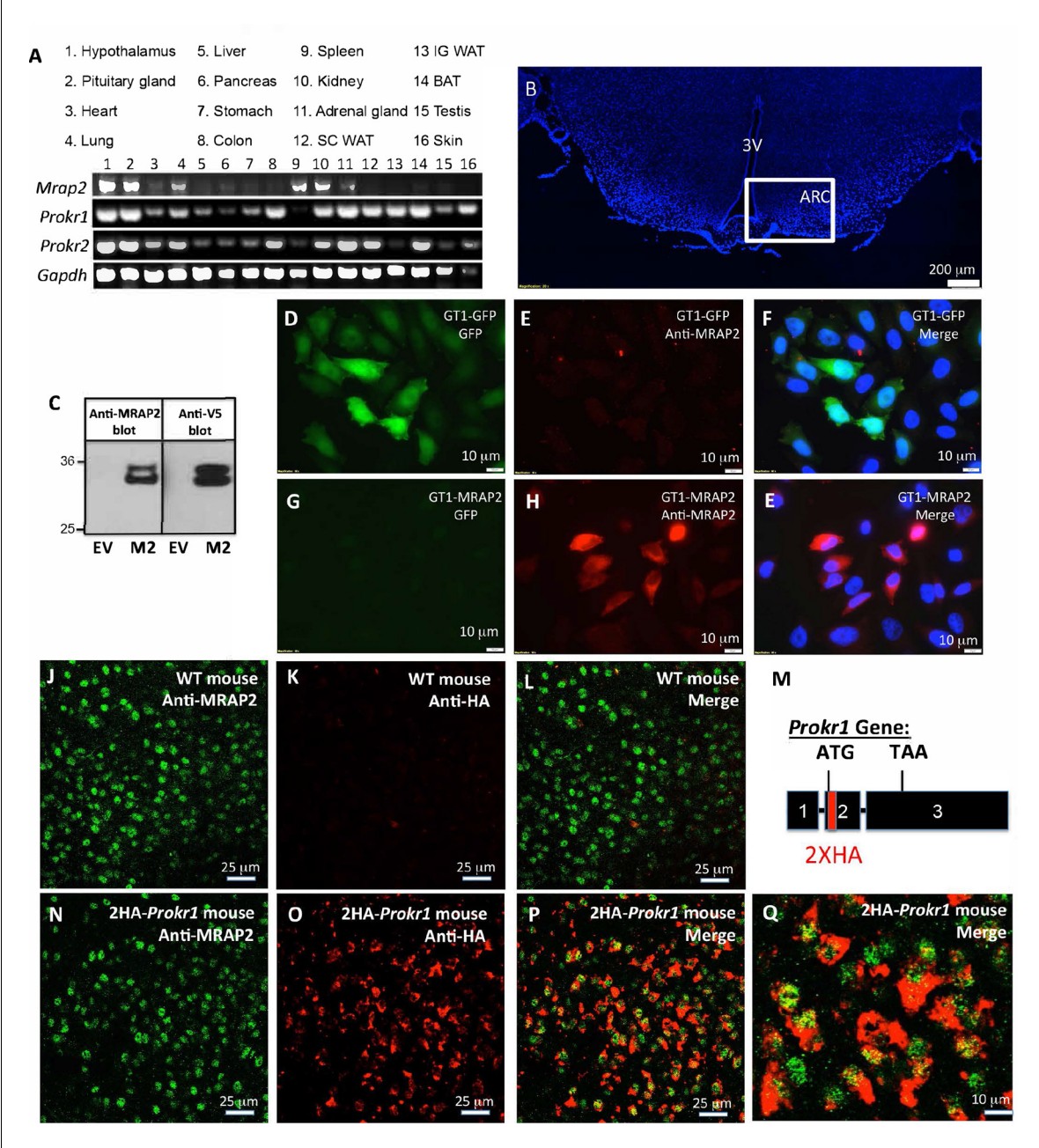

**Figure 1.** Tissue distribution of PKRs and MRAP2. (**A**) Expression of MRAP2, PKR1 and PKR2 mRNA, measured by RT-PCR in tissues harvested from male mice. (**B**) DAPI-stained mouse brain section containing the hypothalamus. Square depicts the position of the arcuate nucleus. (**C**) Validation of anti-MRAP2 antibody by western blot on lysates from CHO cells transfected with empty vector (EV) or mouse MRAP2 (M2). (**D-I**) Validation of the MRAP2 antibody by immunofluorescence using GT1-1 cells stably expressing GFP (**D,E** and **F**) or MRAP2 (**G,H** and **E**). (**J–L**) Confocal images of immunofluorescence detecting MRAP2 (in green) or HA (in red) in the arcuate nucleus of a wild type mouse. (**M**) Schematic representation of the insertion of 2XHA tags after the start codon of the PKR1 gene in mice using CRISPR/Cas9 technology. A more detailed description is depicted in *Figure 1—figure supplement 1*. (**N-Q**) Confocal images of immunofluorescence detecting MRAP2 (in green) or HA-PKR1 (in red) in the arcuate nucleus of the 2HA-*Prokr1* mouse.

The following figure supplement is available for figure 1:

**Figure supplement 1.** Generation of the 2HA-*Prokr1* mouse model.

pituitary gland), the adrenal glands, the lungs, the spleen and the kidneys, but also, at lower level, in the heart and the pancreas (*Figure 1A*). Both PKR1 and PKR2 seem to be expressed in a large number of tissues including brain, heart, lungs, stomach, colon, kidneys, adrenals, fat and testis (*Figure 1A*), thus confirming that MRAP2 and PKRs expression overlap in several organs. Because of the known involvement of both MRAP2 and PKR1 in the regulation of energy homeostasis, and the fact that both PKR1 and MRAP2 mRNA were detected in the hypothalamus, we tested if both proteins co-localized in hypothalamic neurons. In order to detect MRAP2 in brain slices we validated a commercial antibody by western blot (*Figure 1C*) and by immunofluorescence (*Figure 1D–E*). The MRAP2 antibody was validated by western blot using lysates from CHO cells transfected with mouse MRAP2-V5 or empty vector as a control. Both the MRAP2-antibody and the V5-antibody detected the same bands and no signal was detectable in the lysate of mock transfected cells (*Figure 1C*). We also validated the MRAP2 antibody for immunofluorescence using a GT1-1 hypothalamic neuronal cell line stably expressing GFP (GT1-1-GFP) as a control, or MRAP2 (GT1-1-MRAP2). We show that the MRAP2 antibody specifically labeled GT1-1-MRAP2 cells (*Figure 1G,H and E*) but not GT1-1-GFP cells (*Figure 1D, E and F*), further validating the specificity of the antibody. Due to the fact that the cell lines used are not clonal, not all the cells show the same intensity of staining. Unfortunately, the MRAP2 antibody could not be validated on the *Mrap2* KO mouse because of the way this mouse model was generated. Indeed, to produce this mouse, the cassette was inserted in the intron between exon 3 and 4 before the region coding for the transmembrane domain of MRAP2, thus allowing the N-terminal region to be produced (details can be found on the EUCOMM website). This region by itself is soluble and non-functional but can be recognized by the polyclonal MRAP2 antibody. Due to the lack of specific antibody to detect PKR1 in-vivo, we generated a mouse expressing a tagged PKR1. In order for PKR1 expression level, localization and activity to remain unchanged in this mouse model, we used the CRISPR / Cas9 technology to insert 2XHA tag immediately after the start codon of the endogenous PKR1 gene in chromosome 6 (*Figure 1—figure supplement 1* and *Figure 1M*). The insertion of the HA tags was verified by sequencing (*Figure 1—figure supplement 1*). Using immunofluorescence method we show that MRAP2 is readily detectable in cells of the arcuate nucleus in both WT and 2HA-*Prokr1* mice (*Figure 1J and N*). In contrast, PKR1 was only detected, using the HA antibody, in the 2HA-*Prokr1* mouse, therefore validating the specificity of PKR1 staining (*Figure 1K and O*). Finally we show that PKR1 and MRAP2 colocalize in a large number of cells in the arcuate nucleus, making it physiologically relevant to study the regulation of PKR1 by MRAP2 (*Figure 1P and Q*).

To determine if MRAP2 is a regulatory protein of PKR1, we first assessed the ability of those two proteins to form a complex. To this end, we transfected CHO cells with 2HA-PKR1 and MRAP2-3Flag and pulled down either PKR1 with a monoclonal anti-HA antibody or MRAP2 with a monoclonal anti-Flag antibody from the cell lysates. The precipitated proteins were then identified by western blot using an anti-HA antibody to detect the receptor and an anti-flag antibody to detect MRAP2. We observed that a significant fraction of PKR1 co-immunoprecipitates with MRAP2 (*Figure 2A*) and that a large fraction of MRAP2 co-immunoprecipitates with PKR1 (*Figure 2B*), thus demonstrating that PKR1 and MRAP2 interact. Like MRAP1, MRAP2 is detected as two bands, one band representing the non-glycosylated form of MRAP2 and a higher band corresponding to the glycosylated form as confirmed by treatment with the deglycosylation enzyme, PNGase F (*Figure 2E*). In some cases a higher molecular weight smear is detectable but the molecular identity of this band remains unclear. We verified that the bands corresponding to PKR1 in cells transfected with both PKR1 and MRAP2 matched the bands in cells transfected with PKR1 and empty vector (*Figure 2C*), and that the bands corresponding to MRAP2 in cells transfected with both PKR1 and MRAP2 matched the bands in cells transfected with MRAP2 and empty vector (*Figure 2D*) in order to further confirm that the bands observed in the co-immunoprecipitation were in fact PKR1 and MRAP2. We then determined whether this interaction happens in live cells and identified the subcellular localization of the complex. To this end we used a bi-molecular fluorescence complementation assay, or BiFC. PKR1 C-terminally fused to a fragment of YFP (Y2) was co-expressed with MRAP2 C-terminally fused to the complementary fragment of YFP (Y1). Using this method fluorescence can only be achieved if PKR1 and MRAP2 come in close proximity and allow the YFP to complement (*Figure 2F*). We found that, as predicted, expressing only one of the two fusion proteins did not yield any fluorescence (*Figure 2G and H*), whereas YFP fluorescence was readily detectable

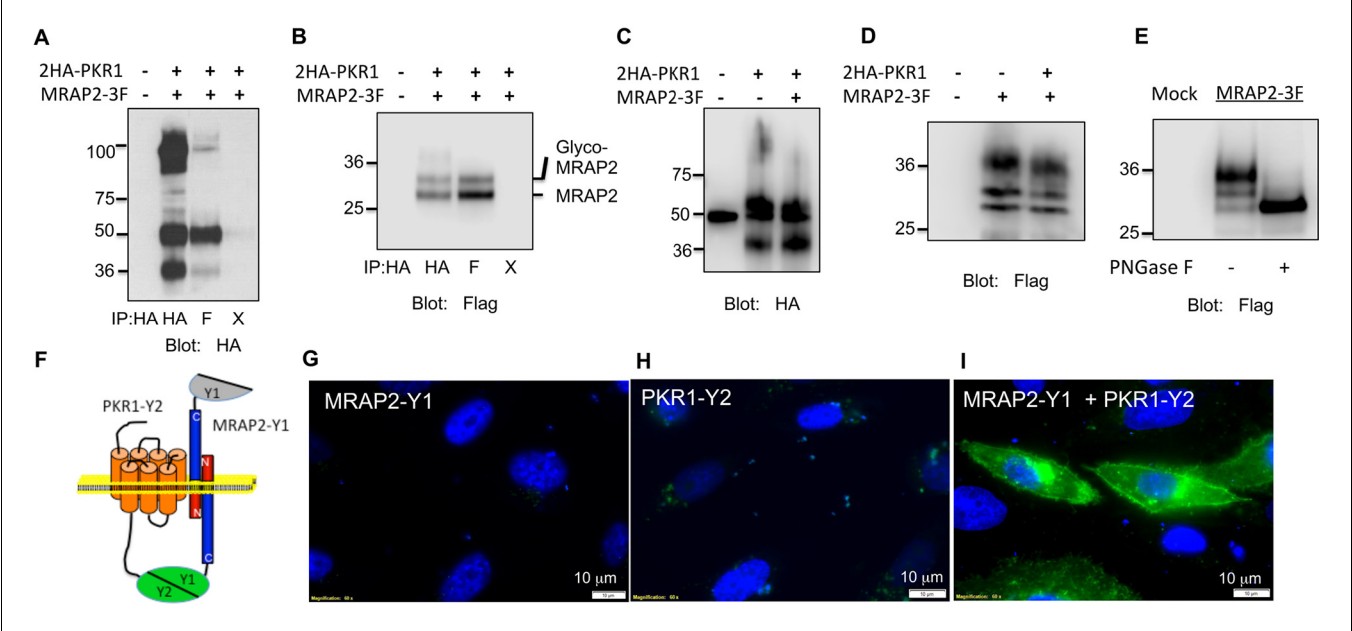

**Figure 2.** PKR1 and MRAP2 coimmunoprecipitation and localization in live cells. (**A**) and (**B**) Co-immunoprecipitation of 2HA-PKR1 and MRAP2-3Flag from transfected CHO cells. PKR1 was detected using rabbit anti-HA antibody (**A**), and MRAP2 was detected using anti-Flag antibody (**B**). X= beads only, no antibody was used for the IP. (**C**) and (**D**) Immunoprecipitation of PKR1 (**C**) or MRAP2 (**D**) from cells expressing PKR1 alone, MRAP2 alone or both. PKR1 was immunoprecipitated and detected with mouse anti-HA and MRAP2 was immunoprecipitated and detected with mouse anti-Flag. (**E**) Western blot detecting MRAP2 treated or not with PNGase F. (**F**) Schematic representation of bimolecular fluorescence complementation (BiFC) between YFP fragments fused to PKR1 and MRAP2. (**G, H** and **E**) CHO cells were transfected with MRAP2-Y1 (**G**), PKR1-Y2 (**H**) or both (**E**) Nuclei stained with Hoechst 33,342 are shown in blue and YFP fluorescence in green.

in intracellular compartments as well as on the plasma membrane of cells expressing both fusion proteins (*Figure 2I*). This result confirms that PKR1 and MRAP2 can interact in live cells.

We then assessed the effect of MRAP2 on PKR1 signaling. Prokineticin receptors are coupled to both the Gαs and the Gαq signaling pathways. In the former they activate adenylyl cyclase, leading to the production of cAMP. In the latter they activate phospholipases, leading to the production of IP3 and to the release of calcium from the ER. In order to measure the effects of MRAP2 on PKRs signaling through the Gαs pathway, CHO cells were transfected with the CRE-luciferase reporter and either PKR1 or PKR2 in the presence or absence of MRAP2. We find that MRAP2 dose-dependently inhibits PKR1 efficacy and decreases its potency four to five fold with a maximum effect at a receptor to MRAP2 DNA ratio between 1:7 and 1:10 (*Figure 3A*). For this reason all subsequent experiments were performed at a 1:10 ratio. To further validate the inhibitory effect of MRAP2 on PKR1, we tested the effect of both human and mouse MRAP2 on cAMP production stimulated by both agonists (PK1 and PK2). Both MRAP2 isoforms inhibited PKR1 responses stimulated by either PK1 (*Figure 3B*) or PK2 (*Figure 3C*) and in both cases mouse MRAP2 was a more effective inhibitor of PKR1 than human MRAP2. In the rest of the study we only use the human MRAP2 since the receptors used are the human isoforms. To rule out the possibility that MRAP2 interfered with the luciferase reporter rather than inhibiting PKR1-mediated cAMP production, we stimulated CHO cells expressing PKR1 with empty vector at a 1:10 ratio or with MRAP2 at the same ratio with increasing concentration of PK2 in the presence of the phosphodiesterase inhibitor IBMX. Accumulated cAMP was measured using the Perkin Elmer LANCE cAMP assay. This assay directly measures cAMP concentration using a TR-FRET technology and does not involve luminescence. MRAP2 inhibited PK2 responses similarly in the two assays (*Figure 3D*), confirming its inhibitory role in PKR1-mediated cAMP signaling.

Because PKR1 can also couple to Gαq, we tested the effect of MRAP2 on PKR1-mediated production of IP3. For those experiments, we used CHO-M1 cells, which stably express the Gαq-coupled muscarinic M1 receptor; this allowed us to use the response to the M1R agonist carbachol to

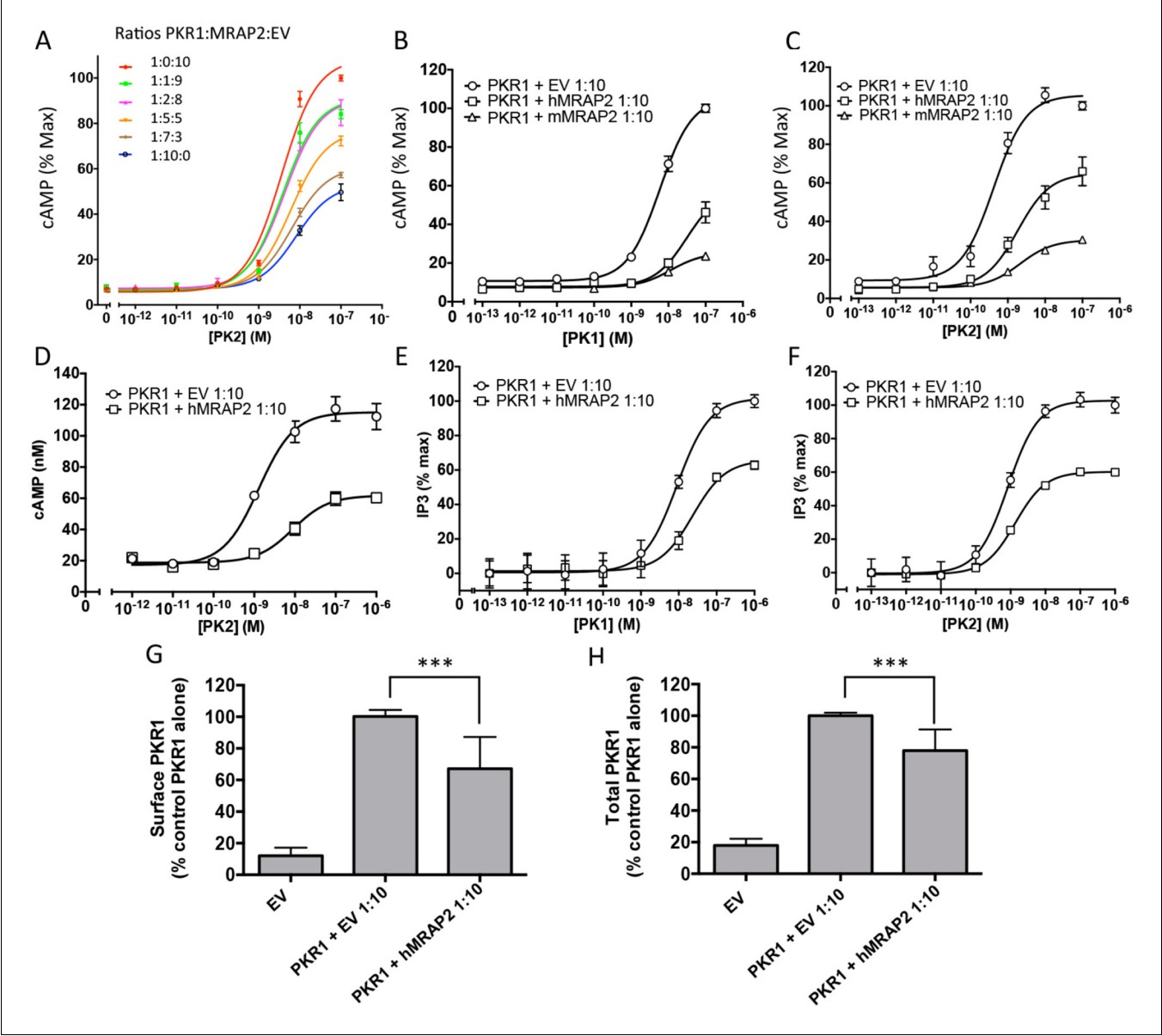

**Figure 3.** MRAP2 inhibits PKR1 signaling. CHO cells in a 10 cm dish were transfected with 2.5 µg CRE-luciferase, 225 ng PKR1 and a total of 2.25 µg of plasmids encoding MRAP2 and/or empty vector in the ratios shown. After 24 hr cells were subcultured for use as follows: (**A**) Cells expressing PKR1 and different amounts of hMRAP2 and/or empty vector were stimulated with PK2 and cAMP responses measured with the CRE-luciferase cAMP assay. (**B** and **C**) Cells expressing PKR1 with empty vector, mouse MRAP2 (mMRAP2) or human MRAP2 (hMRAP2) at a 1:10 ratio were stimulated with (**B**) PK1 or (**C**) PK2 and cAMP responses measured with CRE-luciferase. (**D**). Cells expressing PKR1 with empty vector or hMRAP2 were stimulated with PK2 in the presence of 0.1 mM isobutylmethylxanthine and cAMP concentrations measured with the LANCE cAMP assay. (**E** and **F**) Cells expressing PKR1 with empty vector or mMRAP2 were stimulated with (**E**) PK1 or (**F**) PK2 in the presence of LiCl and IP3 production measured with the IP-One assay. (**G**) Surface or (**H**) total expression of PKR1 in cells transfected with empty vector, 2HA-PKR1 and empty vector, or 2HA-PKR1 and hMRAP2 using cell ELISA assays. One-way ANOVA with Tukey post test *$p<0.05$, **$p<0.01$, ***$p<0.001$

normalize the results and correct for possible differences in growth rates in the different transfection conditions. The assay used measures the accumulation of IP1, a degradation product of IP3, in the presence of lithium, an inhibitor of the inositol monophosphatase. We found that, as with the cAMP pathway, MRAP2 significantly inhibits PKR1-mediated IP3 production in response to both PK1 (*Figure 3E*) and PK2 (*Figure 3F*). This result suggests that MRAP2 can inhibit both known signaling pathways downstream of PKR1.

Since several GPCR accessory proteins have been shown to modulate GPCR trafficking (*Cooray et al., 2009*; *Sexton et al., 2006*; *Metherell et al., 2005*; *Sebag and Hinkle, 2007*; *Matsunami et al., 2009*), we tested the impact of MRAP2 on PKR1 trafficking to the plasma membrane. To this end we measured both the surface density (in non-permeabilized cells) and total expression (in permeabilized cells) of 2HA-PKR1 when expressed with empty vector or with MRAP2 by fixed-cell ELISA. We found that MRAP2 decreased PKR1 expression at the cell surface by approximately 35% (*Figure 3G*). Analysis of total expression revealed that the level of PKR1 is reduced in the presence of MRAP2 by 27% (*Figure 3H*). The fact that the decreases in surface and total expression of PKR1 in the presence of MRAP2 are similar suggests that MRAP2 does not significantly impact PKR1 trafficking.

We also investigated the ability of MRAP2 to regulate PKR2, a GPCR closely related to PKR1. We transfected CHO cells with 2HA- PKR2 and MRAP2-3Flag and performed a co-immunoprecipitation as described earlier. A significant fraction of PKR2 was pulled down with MRAP2 (*Figure 4A*) and conversely, a significant fraction of MRAP2 co-immunoprecipitated with PKR2 (*Figure 4B*). As for PKR1, we confirmed that the bands observed for PKR2 and MRAP2 in the co-immunoprecipitation match the bands obtained with cells expressing either PKR2 (*Figure 4C*) or MRAP2 (*Figure 4D*)

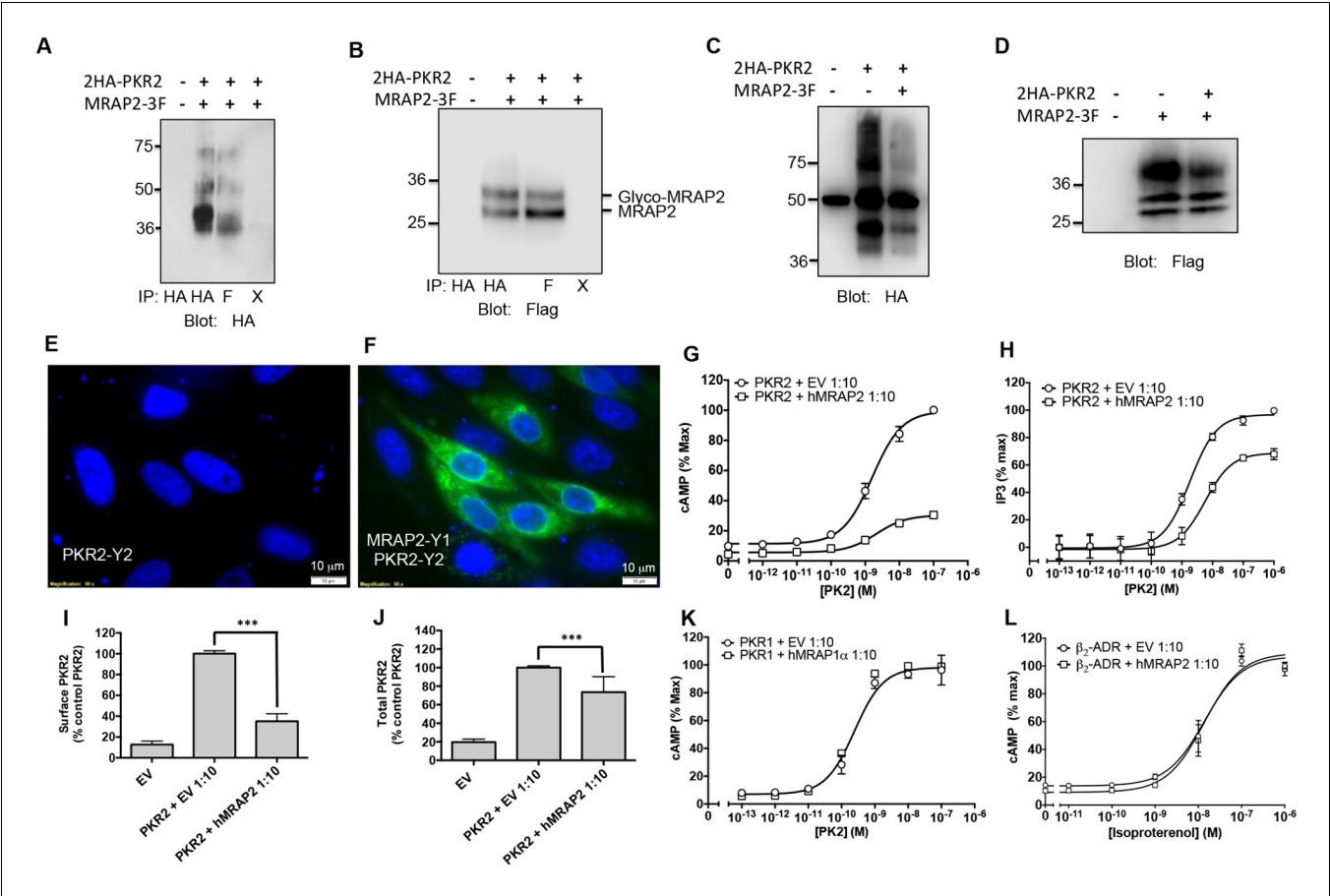

**Figure 4.** Specificity of MRAP2 regulation. (A and B) Co-immunoprecipitation of 2HA-PKR2 and MRAP2-3Flag from transfected CHO cells. PKR2 was detected using rabbit anti-HA antibody (A), and MRAP2 was detected using mouse anti-Flag antibody (B). X= beads only, no antibody was used for the IP. (C and D) Immunoprecipitation of PKR2 (C) or MRAP2 (D) from cells expressing PKR2 alone, MRAP2 alone or both. PKR2 was immunoprecipitated and detected using mouse anti-HA antibody. MRAP2 was immunoprecipitated and detected with mouse anti-Flag). (E and F) CHO cells were transfected with PKR2-Y2 (E) or with PKR2-Y2 and MRAP2-Y1 (F). Nuclei stained with Hoechst 33,342 are shown in blue and YFP fluorescence in green. (G and H) cAMP production stimulated by (G) PK1 or (H) PK2 in cells expressing PKR2 with empty vector or hMRAP2. (I) Surface and (J) total expression of PKR2 in cells transfected with empty vector, 2HA-PKR2 alone or 2HA-PKR2 and MRAP2 using cell ELISA assays. (K) PK2-stimulatedcAMP in cells expressing PKR1 with empty vector or hMRAP1α. (L) Isoproterenol-stimulatedcAMP in cells expressing the $\beta_2$-ADR with empty vector or MRAP2. cAMP responses were measured with the CRE-luciferase assay. One-way ANOVA with Tukey post test *$p<0.05$, **$p<0.01$, ***$p<0.001$

alone. The interaction of PKR2 with MRAP2 was also confirmed by BiFC in cells expressing PKR2 fused to a fragment of YFP and MRAP2 fused to the complementary fragment of YFP. As expected, no YFP fluorescence was detected when only one of the two fusion proteins was expressed (*Figure 2G* and *Figure 4F*), however, YFP fluorescence was readily detectable when both fusion proteins were expressed (*Figure 4F*), especially in intracellular compartments, thus confirming that PKR2 and MRAP2 can form complexes in live cells. We then measured the effect of MRAP2 on PKR2 signaling in response to PK2 through both the cAMP and the IP3 pathway. MRAP2 significantly inhibited PKR2-mediated cAMP and IP3 signaling (*Figure 4G and H*). It is, however, noticeable that MRAP2 inhibited the cAMP and IP3 pathways to the same extent (about 50%) with PKR1 (*Figure 3B, C, E and F*), whereas MRAP2 inhibited the cAMP pathway more effectively than the IP3 pathway downstream of PKR2 (about 70% for cAMP vs. about 20% for IP3) (*Figure 4G and H*). Unlike for PKR1, MRAP2 decreased PKR2 efficacy but did not significantly affect its potency. We also measured the effect of MRAP2 on PKR2 trafficking and showed that, unlike with PKR1, MRAP2 decreased the surface expression of PKR2 by over 70% (*Figure 4I*) with only 20% to 30% change in total receptor expression (*Figure 4J*), suggesting a possible effect of MRAP2 on PKR2 trafficking. In fact, in the case of PKR2, the extent of the trafficking inhibition is comparable to the inhibition in cAMP signaling.

It is clear that MRAP2 can regulate multiple GPCRs, including PKRs, MC4R (*Sebag et al., 2013*; *Asai et al., 2013*) and MC5R (*Sebag and Hinkle, 2009a*; *Chan et al., 2009*). Nonetheless, MRAP proteins are highly selective for their targets since MRAP1 did not modulate PKR1 signaling (*Figure 4K*) and MRAP2 did not affect β2-adrenergic receptor signaling (*Figure 4L*). We previously reported that MRAP2 does not modify signaling by several other GPCRs involved in the regulation of energy homeostasis including the Y2 receptor, Y1 receptor, glucagon-like peptide 1 receptor and the MC3R (*Sebag et al., 2013*).

To further validate our findings, we assessed the regulatory action of MRAP2 on PKR1 and PKR2 signaling (*Figure 5A and B*) and trafficking (*Figure 5C and D*) in a more relevant cell line, the hypothalamic neuronal GT1-1 cell line, and found very similar results to those obtained in CHO cells.

We then investigated if MRAP2 regulates PKR1 activity in-vivo in mice. Male and female WT and *Mrap2* KO sibling mice were cannulated in the lateral ventricle of the brain. After recovering from the surgery, mice were fasted overnight and injected ICV with vehicle or the indicated dose of PK2 10 min prior to being given access to food. Food intake was then measured at 30 min, 1, 2, 4 and 6 hr. As expected the highest dose of PK2 significantly inhibited food intake in both WT and *Mrap2* KO mice, however, the anorexigenic effect was more pronounced in *Mrap2* KO mice (*Figure 6*). This is especially evident at the lowest PK2 dose injected since this dose has no significant effect in both male and female WT mice but inhibits up to 70% of food intake in *Mrap2* KO mice (*Figure 6*). The effect of PK2 seemed to be more potent and longer lasting in female compared to male *Mrap2* KO mice. These results confirm that MRAP2 significantly inhibits the anorexigenic signal mediated by PKR1 in vivo. Both in published studies and in our hands, PK2 injections decreased food intake without causing malaise or sickness like behavior in the animals (*Gardiner et al., 2010*; *Beale et al., 2013*).

Because MRAP2 has been shown to regulate the MC4R (*Sebag et al., 2013*) and that it had been suggested that the anorexigenic effect of PKR1 may be working upstream of the MC4R (*Gardiner et al., 2010*), we tested if MC4R signaling was required for PKR1 stimulation to decrease food intake. To this end we injected overnight fasted WT and *Mc4r* KO mice ICV with vehicle or 0.65 µg PK2 before giving them access to food. We show that, like for WT mice, PK2 caused a very significant decrease in food intake over 6 hr in both male and female *Mc4r* KO mice (*Figure 7*), demonstrating that PKR1 can inhibit food intake independently of the MC4R (*Figure 6*). Like observed in the previous experiment PK2 seems to be more efficacious in female mice. The difference in food intake between *Mc4r* KO and WT mice injected with saline may be attributed to the documented enhanced stress-induced anorexia in *Mc4r* KO mice (*Liu et al., 2007*; *Vergoni et al., 1999*). Additionally, a more potent anorexigenic effect of PK2 was measured in the *Mc4r* KO colony compared to the *Mrap2* KO colony, possibly due to slightly different genetic backgrounds.

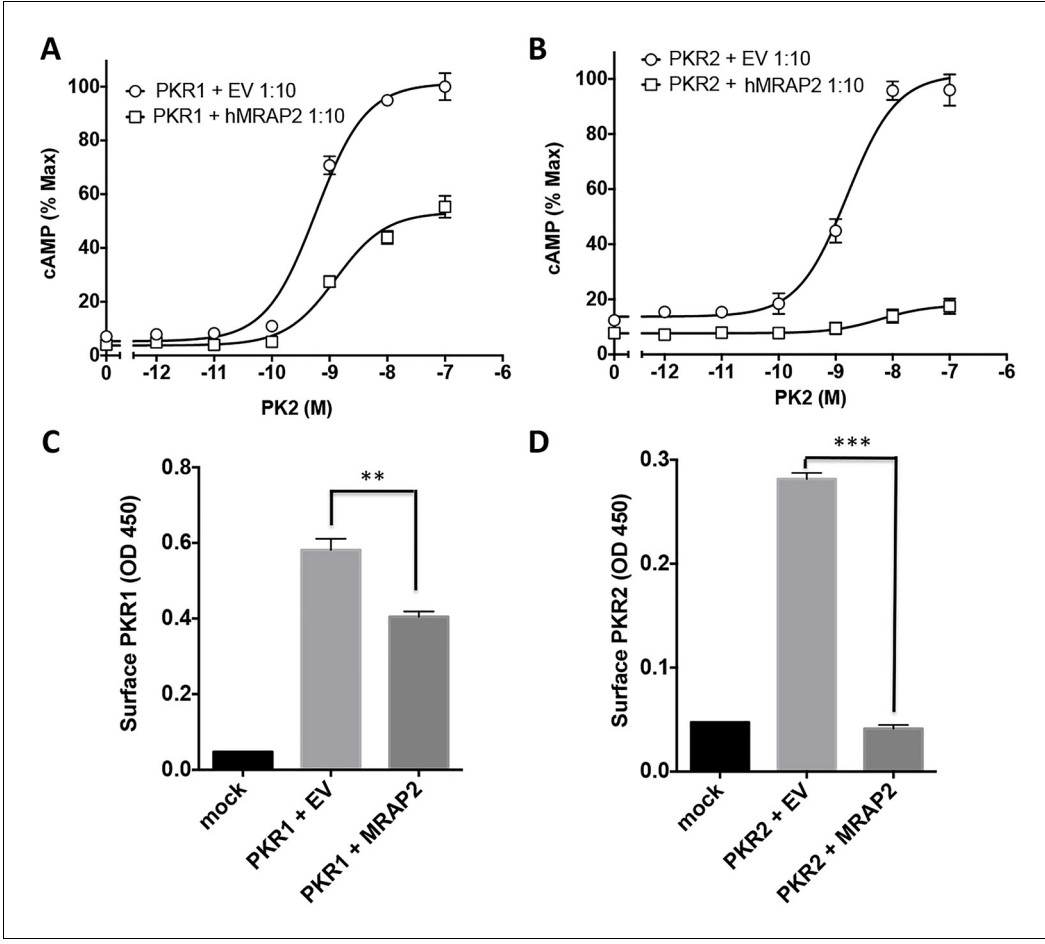

**Figure 5.** Regulation of PKR1 and PKR2 signaling and trafficking in GT1-1 cells. (**A**) GT1-1cells expressing CRE-Luciferase, 2HA-PKR1 and either empty vector or hMRAP2 were stimulated with PK2 and cAMP responses measured with the CRE-luciferase cAMP assay. (**B**) Cells expressing CRE-Luciferase, 2HA-PKR2 and either empty vector or were stimulated with PK2 and cAMP responses measured with CRE-luciferase. (**C** and **D**) ELISA measuring the surface expression of PKR1 (**C**) or PKR2 (**D**) in GT1-1 cells in the presence or absence of MRAP2. One-way ANOVA with Tukey post test *$p<0.05$, **$p<0.01$, ***$p<0.001$

## Discussion

MRAP proteins are known to interact with all five melanocortin receptors and to regulate the trafficking and/or signaling of MC2R, MC4R and MC5R (*Metherell et al., 2005*; *Sebag and Hinkle, 2007*; *2009a*; *Sebag et al., 2013*; *Cerdá-Reverter et al., 2013*; *Asai et al., 2013*). In this study we identify a non-melanocortin GPCR regulated by a member of the MRAP protein family, i.e. the prokineticin receptors. This finding not only demonstrates the importance of considering the expression of MRAP2 when studying PKRs, but also establishes that MRAPs, especially MRAP2 due to its broader tissue distribution, can regulate various GPCRs and are involved in numerous physiological functions. We also show that, unlike all the other GPCR accessory proteins (RAMPs, REEP and RTPs), which to date have been shown only to promote GPCR trafficking or signaling, MRAPs can either potentiate or inhibit the GPCRs they regulate. For example, whereas MRAP1 is required for MC2R activity, it prevents MC5R trafficking and homodimerization (*Sebag and Hinkle, 2009a*). In zebrafish, the two isoforms of MRAP2, zMRAP2a and zMRAP2b, have opposite effects on MC4R signaling (*Sebag et al., 2013*) and here we demonstrate that mammalian MRAP2, which had previously been shown to potentiate MC4R signaling (*Asai et al., 2013*), strongly inhibits PKR1 and PKR2 signaling.

GPCR accessory proteins have so far been described as potentiators of GPCR trafficking and/or signaling (*Ritter and Hall, 2009*) with the exception of MRAPs. Indeed, to our knowledge, the ability

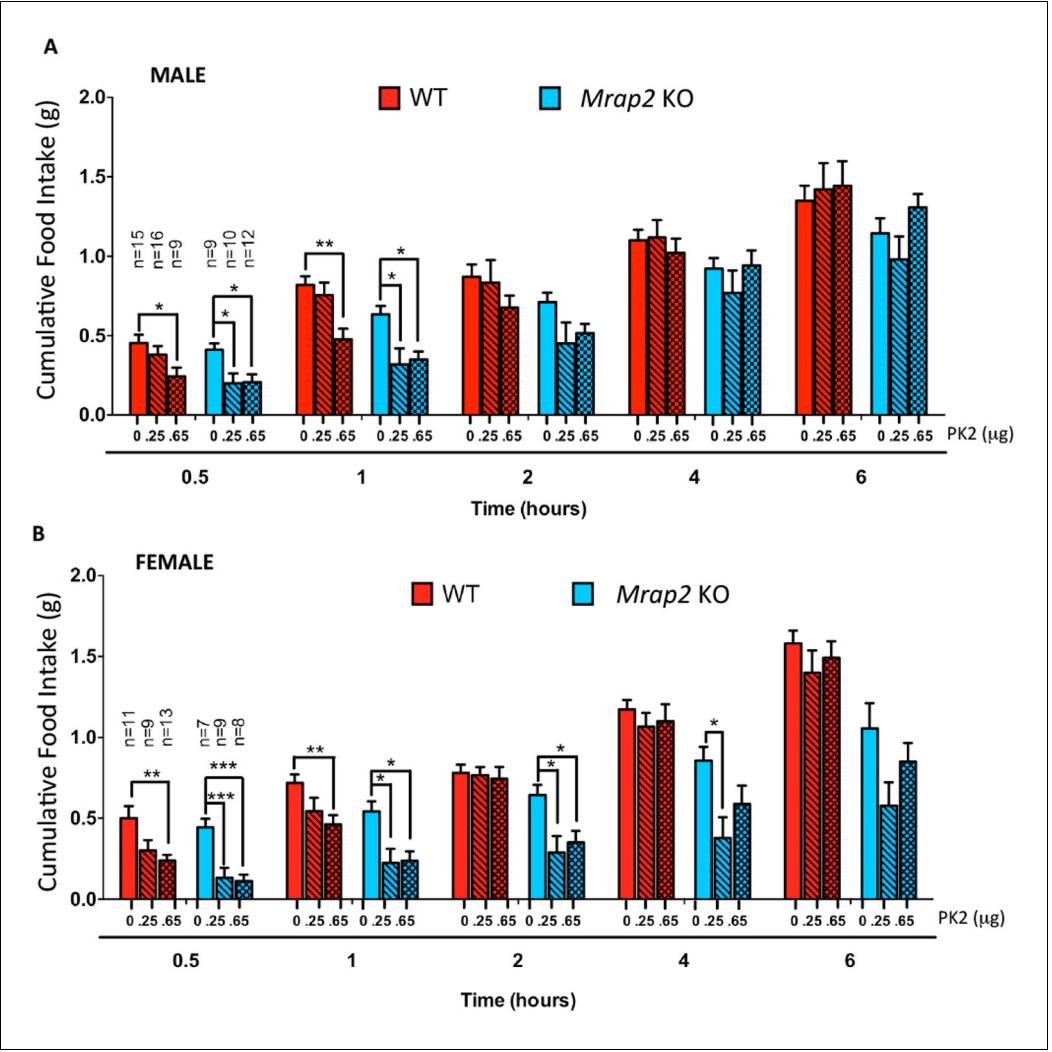

**Figure 6.** MRAP2 mediated regulation of PKR1 anorexigenic effect. Cumulative food intake in male (**A**) and female (**B**) WT and *Mrap2* KO mice after overnight fast and ICV injection of saline or the indicated dose of PK2. One-way ANOVA with Tukey post test *p<0.05, **p<0.01, ***p<0.001.

to selectively potentiate a subset of GPCRs while inhibiting another is unique to MRAP proteins. As such, it will be exciting to pursue the study of the mechanisms underlying MRAP2 targeting of specific GPCRs and the directionality of the resulting regulation.

The finding that MRAP2 inhibits rather than activates PKRs was unexpected since both PKRs and MRAP2 are thought to promote leanness. In fact, knocking out PKR1 (*Szatkowski et al., 2013*) or MRAP2 (*Asai et al., 2013*) in mice causes obesity, and central administration of the PKR1 agonist PK2 leads to a significant decrease in food intake (*Gardiner et al., 2010*; *Beale et al., 2013*) and *Figures 6* and *7*. It is, however, important to note that Asai et al. showed that MC4R / MRAP2 double KO mice have an intermediate obesity phenotype between the *Mrap2* KO and the *Mc4r* KO mice (*Asai et al., 2013*). In other words, losing the function of both MRAP2 and MC4R causes a milder obesity than losing the function of MC4R alone (*Asai et al., 2013*), suggesting that MRAP2 can promote weight gain through a mechanism that is distinct and independent from the regulation of the MC4R. Further arguing this hypothesis, we showed that the anorexigenic effect of PKR1 is independent of MC4R since central injection of PK2 in fasted *Mc4r* KO mice significantly decreased food intake. Consequently, we conclude that MRAP2-mediated inhibition of PKR1 is an important mechanism through which MRAP2 promotes hunger, energy intake and weight gain and that MRAP2 is a major component of the machinery that controls energy homeostasis. The cause of the

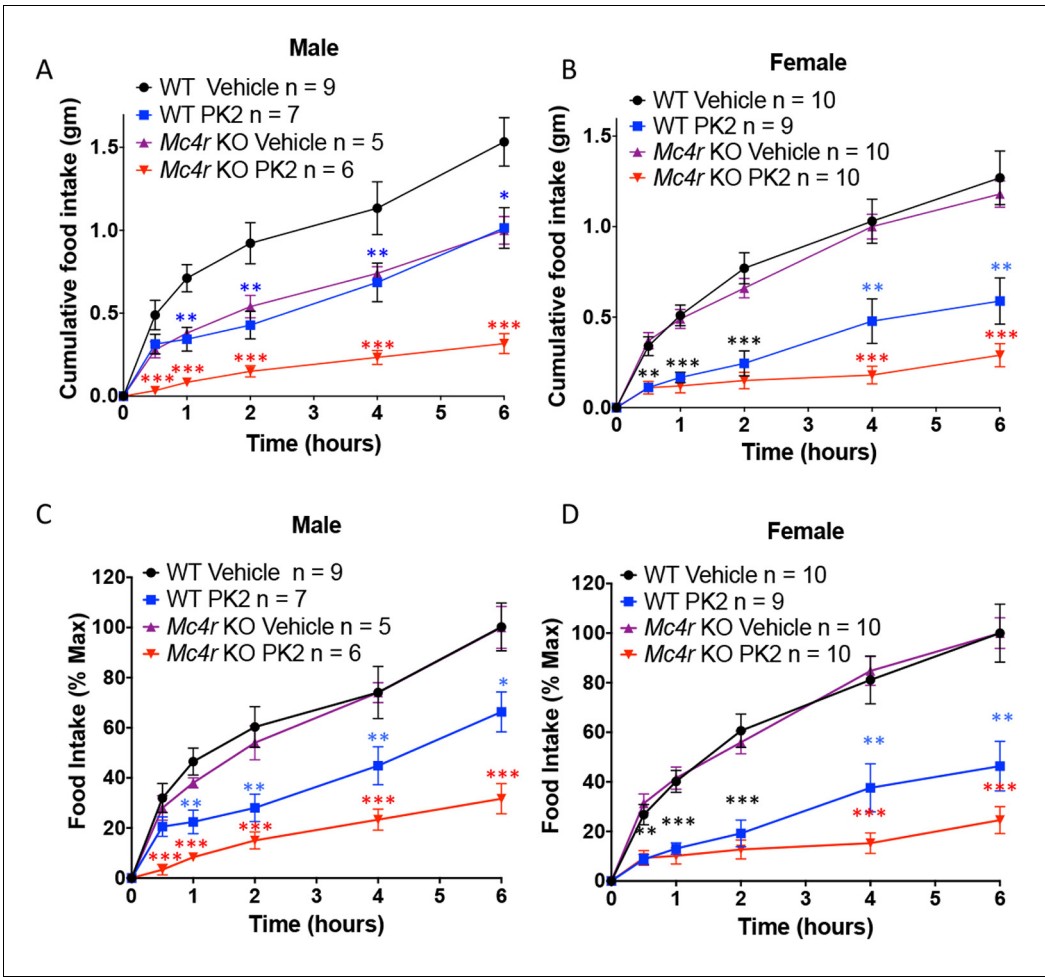

**Figure 7.** The Anorexigenic effect of PKR1 does not require MC4R. Cumulative food intake in male (**A**) and female (**B**) WT and *Mc4r* KO mice after overnight fast and ICV injection of saline or the indicated dose of PK2. (**C** and **D**) Same results as those depicted in A and B but normalized to the food intake of the 'Vehicle' injected mice of the same genotype at the 6 hr time point. T-test *p<0.05, **p<0.01, ***p<0.001.

more potent anorexigenic effect of PK2 in female compared to male *Mrap2* KO mice remains unclear and will require further studies. It is also important to note that PKR1 and PKR2 are expressed in many tissues and play roles in numerous physiological processes for which negative regulation by MRAP2 could be critical.

Increasingly, GPCRs are found to be regulated by accessory proteins, thus strongly suggesting that the importance of this mode of receptor regulation is underestimated. Moreover, our study indicates that accessory proteins not only potentiate, but can also inhibit receptor signaling. For these reasons, identifying the accessory proteins that are associated with specific GPCRs is an important step in accurately determining their pharmacology. Additionally, a better understanding of GPCR regulation by accessory proteins will allow the development of more accurate and more relevant model systems for the identification of small molecules targeting receptors and will increase the likelihood of such compounds to retain their activity in vivo. Finally, this study illustrates the important role of the interaction of PKR1 and MRAP2 in the regulation of energy homeostasis and suggests that this complex may be a valuable new target for the treatment of obesity.

# Materials and methods

### Animals

*Mrap2* KO mice are on a C57BL/6 background and were obtained from the EUCOMM program. LoxTB *Mc4r* KO mice were a generous gift from Dr. Lutter, University of Iowa. Both male and female mice were used and all experiments using mice were approved by the animal care and use committee at the University of Iowa. The 2HA-*Prokr1* mouse was generated by the genome editing core facility of the University of Iowa using pronuclear injection of a guide RNA targeting the first exon of the *Prokr1* gene, Cas9 and a repair DNA containing the sequence of 2XHA tags immediately after the start codon. Mice were screened for the insertion by PCR and validated by sequencing.

### Plasmids and antibodies

The *Prokr1* and *Prokr2* plasmids, encoding PKR1 and PKR2 respectively, were obtained from the Missouri S&T cDNA Resource Center. The N-terminal 2HA tag was added to *Prokr1* and *Prokr2* by PCR using the *Prokr1* and *Prokr2* plasmids as templates and the tagged receptor gene inserts were cloned into pcDNA5. h*Mrap2* plasmid was a generous gift from Dr. Patricia Hinkle, University of Rochester. Plasmids encoding the split-YFP were a kind gift from Dr. Catherine Berlot, Weis Center for Research, Geisinger Clinic, Danville, PA. *Mrap2, Prokr1* and *Prokr2* were amplified and inserted, in frame, 5' of the YFP fragments. The plasmid encoding CRE-*luc* was a kind gift from Dr. George Holz, New York University School of Medicine, New York, NY. The validity of all constructs was verified by sequencing.

The antibodies used in this study were mouse monoclonal anti-HA (HA11) (Biolegend, San Diego, CA), rabbit anti-HA (cell signaling, Danvers, MA), mouse monoclonal anti-V5 (AbDSerotec, Raleigh, NC), M2 anti-Flag (Sigma-Aldrich, St. Louis, MO), horseradish peroxidase (HRP)-conjugated antibodies against mouse and rabbit immunoglobulin (Biorad, Hercules, CA) and anti-MRAP2 rabbit polyclonal antibody (Novus Biological, Littleton, CO).

### Cell cultures and transfections

HEK293T and CHO-K1 were purchased from ATCC, CHO-M1 cells (stably expressing the M1 muscarinic receptor) were obtained from Molecular Devices, GT1-1 cells were kindly provided by Dr. Richard Weiner, University of California San Francisco School of Medicine. All cell lines are mycoplasma free.

HEK293T, CHO-K1, CHO-M1 and GT1-1 cells were cultured in Dulbecco's Modified Eagle's Medium (DMEM)/F-12 supplemented with 5% fetal bovine serum and 1% penicillin-streptomycin, in a humidified atmosphere consisting of 5% $CO_2$ at 37°C. For transfection, cells were grown to 70% confluency. Transfections were performed using LipoD293 in vitro transfection reagent (Signagen, Rockville, MD) for all cells except GT1-1 cell that were transfected with Lipofectamine 3000. Total plasmid concentration was kept identical for all transfections by the addition of empty vector.

### Co-Immunoprecipitation and western blot

HEK293T cells transfected with the indicated plasmids were lysed in 0.1% *n*-dodecyl-β-maltoside in PBS with protease inhibitors. Lysates were centrifuged and supernatants were incubated with the indicated antibody (mouse anti-HA or mouse anti-Flag M2 ) at 1/5000 dilution overnight at 4°C. Immune complexes were collected with protein-G Dynabeads (Life Technologies, Carlsbad, CA) at 4°C for 1 hr. Beads were washed three times and resuspended in LDS loading buffer with 5% β-mercaptoethanol and boiled 5 min. Proteins were resolved by SDS/PAGE and detected by western blot using rabbit anti-HA for PKR1 and PKR2 coimmunoprecipitation experiments, mouse anti-HA or anti-Flag for all other experiments. For MRAP2 deglycosylation, samples were incubated with PNGaseF for 1h at 37°C after the immunoprecipitation step.

### RT-PCR

Tissues were harvested from mice and mRNA was extracted using Trizol. Reverse transcription was carried out using 1 μg of mRNA per sample using the iScript kit (Biorad, Hercules, CA). *Prokr1, Prokr2, Mrap2* and *Gapdh* were then amplified using the prepared cDNA as template and the

following primers: m*Prokr1* Forward: 5' CAC CAA CTT GCT TAT CGC CAA CC 3'; m*Prokr1* Reverse: 5' GGC CAG ATC TGA CCA CAG AAG AT 3'; m*Prokr2* Forward: 5' TGG CCA TCT CTG ACT TCC TGG T 3'; m*Prokr2* reverse:: 5' TAG GAT TTG TAG TAG AGC TGC TGG T 3'; m*Mrap2* Forward: 5'TGT AAA GCC TGC GGT AAC CC 3'; m*Mrap2* Reverse: 5' AGG ACT CCG CGT TGT CTT G 3'; *Gapdh* Forward: 5' GGA GAG TGT TTG CTC GTC CC 3' and *Gapdh* Reverse: 5' ACT GTG CCG TTG AAT TTG CC 3'.

## cAMP assay (Luminescence)

CHO-K1 cells were plated in 10 cm dishes. The next day, they were transiently transfected with a plasmid encoding the CRE-luciferase reporter (firefly luciferase driven by 4 repeats of a non-palindromic cAMP responsive element), along with PKR1 or PKR2 and either empty vector or MRAP2 (1:10 ratio receptor to MRAP2). 24 hr following transfection, the cells were transferred to a white 96-well plate and left to adhere overnight. Cells were then incubated with vehicle or varying concentrations of peptide agonist (PK1 or PK2) or 20 μM forskolin, in DMEM/F-12 supplemented with 0.1% bovine serum albumin (BSA) for 4 hr at 37°C. The medium was removed and 100 μl of luciferin in lysis buffer, 200 mM Tris-HCl, 10 mM MgCl$_2$, 300 uM ATP, 1% Igepal, protease inhibitor cocktail (Roche, Basel, Switzerland), 12.2 mM Acetyl Coenzyme A; 30 μg/ml luciferin (Goldbio, Olivette, MO), was added. Samples were incubated at room temperature for 5 min and luminescence was then measured using a Spectramax I3 plate reader (Molecular Devices, Sunnyvale, CA). Because CRE drives the expression of luciferase, the level of expression of the enzyme is proportional to the cAMP produced in the cell and luminescence serves as a reporter of cAMP production. In addition to the CRE-luciferase reporter, cells were transfected with a constant amount of PKR1 and increasing concentrations of MRAP2 plus enough empty vector to maintain a constant total DNA concentration. Cells were then treated with vehicle or with the indicated concentration of PK2 prior to measuring the cAMP production. In order to correct for possible variations in transfection efficiencies, the results were normalized to the signal obtained in response to 20 μM forskolin in each condition. The results were then normalized to the highest signal obtained in cells expressing PKR1 and CRE-luc in the absence of MRAP2.

## cAMP assay (TR-FRET competition assay)

CHO cells in 10 cm dishes were transfected with indicated plasmids. The next day, cells were plated in a white 96 well plate and allowed to adhere overnight. Cells were then incubated with agonist at the indicated concentration in the presence of 0.1 mM 3-isobutyl-1-methylxanthine for 30 min at 37°C. Cells were then processed using the LANCE cAMP assay kit (PerkinElmer, Waltham, MA) following manufacturer's manual. Plate was read with a Spectramax i3 plate reader equipped with a HTRF (homogeneous time resolved fluorescence) module.

## Fixed cell enzyme-linked immunosorbent assay (ELISA)

Cells were grown in a 24-well plate and transiently transfected with empty vector (mock), with 2HA-PKR1 or 2HA-PKR2 and empty vector, or with 2HA-PKR1 or 2HA-PKR2 and MRAP2. 24 hr after transfection, the cells were washed with PBS, fixed for 10 min with 4% paraformaldehyde in PBS, washed, blocked in either 5% milk in PBS (surface-expression assay) or 5% milk in RIPA buffer (150 mM NaCl, 50 mM Tris, 1 mM EDTA, 1% Triton X-100, 0.1% SDS, 0.5% sodium deoxycholate, pH 8.0) (total-expression assay). Cells were then incubated with 1/5000 anti-HA antibody in blocking buffer for 2 hr at RT, washed three times for 5 min in PBS, incubated with 1/5000 anti-mouse-HRP antibody for 1 hr RT, and washed three times with PBS for 5 min. ELISA substrate (3,3',5,5'-Tetramethylbenzidine, Sigma-Aldrich) was then added until blue color was visible, and the reaction was stopped with 10% sulfuric acid. Absorbance was measured at 450 nm using a Spectramax I3 plate reader.

## Bimolecular fluorescence complementation assay (BiFC)

CHO-K1 cells were seeded on sterilized coverslips and transfected with MRAP2-Y1, PKR1-Y2 or PKR2-Y2 plus empty vector, or PKR1-Y2 or PKR2-Y2 plus MRAP2-Y1. The next day, cells were incubated at room temperature for 1 hr and nuclei were stained with 1 μg/mL Hoechst 33,342 (Life Technologies) for 15 min. Coverslips were then mounted in imaging chambers, and imaging was performed using a 60X objective with an Olympus IX83 inverted fluorescence microscope.

## Inositol phosphate assay

Inositol phosphate assay was performed using the IP-One-Tb HTRF kit (Cisbio Bioassays, France) following the manufacturer's instructions. CHO-M1 cells were transfected with PKR1 or PKR2 plus either empty vector or MRAP2 at a ratio of 1:10 (receptor to MRAP2). The day before the experiment, the transfected cells were plated in a white 384-well plate at a density of 20,000 cells/well. The next day the cells were treated with different concentrations of PK1, PK2 or 1 µM carbachol in stimulation buffer containing lithium for 1 hr at 37°C, and then incubated with IP1-d2 conjugate and Anti-IP1 cryptate Tb in lysis buffer for 1 hr at room temperature. HTRF signal was read at 615 nm and 665 nm using a Spectramax I3 plate reader equipped with the HTRF Cisbio cassette. For each condition, signal obtained with 1 µM carbachol, an agonist of the M1 muscarinic receptor, was used for normalization. Results were then normalized to the highest signal from the control (receptor + empty vector).

## Double fluorescence immunofluorescence

2HA-*Prokr1* mice and wild type littermates were deeply anesthetized with isoflurane and perfused transcardially with PBS and subsequently with 4% PFA in PBS. The whole brains were dissected out and post-fixed in 4% PFA overnight at 4°C. The brains were put in 30% sucrose in PBS until sunk. 40 µm frozen brain sections were cut by a cryostat and treated with 0.4% Triton X-100 in PBS for 1 hr, subsequently incubated in 3% hydrogen peroxide in PBS for 30 min and rinsed two times with PBS before incubation in 10 mM sodium citrate at 80°C for 30 min. The sections were blocked in 5% normal goat serum in 0.4% Triton X-100 in PBS and then in Mouse on mouse (MOM) Ig block reagent at room temperature for 1 hr according to the manufacturing protocol (Vector Laboratories, Burlingame, CA), rinsed with PBS twice, further incubated in MOM diluent for 5 min and subsequently incubated in 1:300 mouse anti-HA antibody in MOM diluent for 30 min. The free floating sections were washed twice with PBS and incubated in 1:250 biotinylated anti mouse IgG antibody for 10 min, washed with PBS three times and incubated in 1:1000 MRAP2 rabbit antibody in 0.4% Triton X-100 in PBS supplemented with 5% normal goat serum overnight at 4°C. The sections were washed with 0.4% Triton X-100 in PBS three times and incubated in 1:250 Pierce high sensitivity streptavidin-HRP (Cat. No. 21130, Thermo Scientific) mixed with 1:250 Alex Fluor 546 goat anti rabbit antibody for 1 hr in 5% normal goat serum, 0.4% triton X-100 in PBS. The sections were washed four times and incubated in 1:100 Alexa Fluor 647 tyramide in amplification buffer for 10 min according to the manufacturer protocol (Cat. No. T20926, molecular probes life technologies) and washed with 0.4% Triton X-100 in PBS four times. The sections were mounted on a glass slide and air-dried. Prolong diamond antifade mountant with DAPI was applied on the sections before images were acquired with a Leica SP8 STED confocal microscope.

For immunofluorescence in cells, GT1-1-GFP and GT1-1-MRAP2 cells were plated on sterile coverslips, fixed in 4% PFA, blocked in 5% goat serum in PBS supplemented with 0.3% Triton X-100 for 1 hr, then incubated with anti-MRAP2 antibody at 1/1000 dilution in blocking buffer for 2 hr at room temperature, washed with PBS, incubated with anti-rabbit-alexa 546 sary antibody for 1 hr, washed with PBS. The coverslips were then mounted on slides with prolong diamond with dapi and imaged.

## Effect of PK2 on food intake

Individually housed male and female MRAP2 knockout mice and sibling WT or loxTB *Mc4r* KO and sibling WT at age 6–7 week-old were anesthetized with 17.5 mg/ml ketamine/ 2.5 mg/ml xylazine mix at dose 0.1 ml per 20 g body weight. A stainless steel guide cannula (Plastics One, USA) was stereotaxically placed into the lateral ventricle (1 mm lateral, 0.3 mm posterior and 2.5 mm ventral from bregma). Perioperatively, mice were administered subcutaneously 0.1 mg/kg buprenorphine. Animals were kept in a temperature and humidity controlled room under a 12 hr light/dark cycle and allowed to recover for seven days after surgery and were handled for five days before the start of the experiment. Animals were fasted overnight and injected ICV with vehicle, 0.25 µg or 0.65 µg PK2 in 3 µl saline 10 min before food was returned to the cages (only vehicle or 0.65 µg PK2 were used for the *Mc4r* KO experiments). Food in each cage was weighed at 30 min, 1, 2, 4 and 6 hr following re-feeding. A minimum of three days drug-free period was maintained between infusions. During this period, the animals were handled but not tested. The mice that had received PK2 at the

first injection received saline at the second and the mice that received saline the first time received PK2.

The cannula placement was verified using histological methods. Animals were euthanized with carbon dioxide and received an ICV injection with 2 μl of ink. Brains were removed and the lateral ventricles opened to check for ink staining. The minimal number of number of animals to be used was determined by power analysis.

## Statistical analyses

All experiments were repeated separately a minimum of three times. Statistics were calculated as follow. For ELISA histograms, one-way ANOVA with Tukey post-test was used and significance was measured between the control (no MRAP2) and the test (with MRAP2). For food intake studies in WT and *Mrap2* KO mice, statistics were measured using a one way ANOVA within the result for the different doses of PK2 at a single time point and for one genotype and one gender. Lines connect the results compared for which the statistical significance is noted. For food intake studies in WT and *Mc4r* KO mice, statistics were measured a T-test between the result for vehicle or PK2 injected within the same gender and same time point, asterisk were color coded for clarity. $*p<0.05$, $**p<0.01$ and $***p<0.001$. Results are shown as mean ± SEM.

## Acknowledgements

Funding was received from the Fraternal Order of Eagle Diabetes Research Center and a Carver Trust Young Investigator Award. We thank the University of Iowa Genome Editing Facility for the generation of the 2HA-*Prokr1* mouse model.

## Additional information

### Funding

| Funder | Author |
| --- | --- |
| Fraternal Order of Eagles Diabetes Research Center | Julien A Sebag |
| Carver Trust Young Investigator award | Julien A Sebag |

The funders had no role in study design, data collection and interpretation, or the decision to submit the work for publication.

### Author contributions

ALC, DS, JAS, Conception and design, Acquisition of data, Analysis and interpretation of data, Drafting or revising the article; EEG, Performed several key experiments included in this manuscript, Acquisition of data, Analysis and interpretation of data, Drafting or revising the article

### Author ORCIDs

Julien A Sebag, http://orcid.org/0000-0002-8366-312X

### Ethics

Animal experimentation: This study was performed in strict accordance with the recommendations in the Guide for the Care and Use of Laboratory Animals of the National Institutes of Health. All of the animals were handled according to approved institutional animal care and use committee (IACUC) protocols (# 4061063 (Sebag)) of the University of Iowa. The protocol was approved by the Office of the IACUC at the University of Iowa. All surgery was performed under ketamine/ xylazine anesthesia, and every effort was made to minimize suffering including the use of post surgery buprenorphine.

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
