## [Decision Letter]

Thank you for submitting your work entitled "The Melanocortin Receptor Accessory Protein 2 Promotes Food Intake Through Inhibition of the Prokineticin Receptor-1" for consideration by *eLife*. Your article has been favorably evaluated by a Senior Editor and three reviewers, one of whom is a member of our Board of Reviewing Editors. The reviewers have discussed the reviews with one another and the Reviewing Editor has drafted this decision to help you prepare a revised submission.

Summary:

Chaly and colleagues describe results from a series of studies investigating the functional interactions of melanocortin receptor accessory protein 2 (MRAP2) and the prokineticin receptor 1 (PKR1). The manuscript provides a strong series of in vitro experiments, all properly controlled, that MRAP2 directly interacts with PKR1 and PKR2 and MRAP2 inhibits the activity of the prokineticin receptors through both Gs and Gq signaling pathways. There was little evidence for altered trafficking of PKR1 to the cell surface, while MRAP2 decreased the cell surface expression of PKR2. These are novel and potentially important findings as they demonstrate interaction of MRAP2 with a non-melanocortin receptor, and they show inhibition of PKR signaling instead of activation, as with the melanocortin receptors. The studies are well described and performed and provide novel information that may be of wide interest. A few issues need to be addressed.

Essential revisions:

1) The majority of the signaling and interaction data is done using heterologous cell systems. While this is clearly useful and understandable the in vivo data is less developed. They report extensive co-expression of the PKRs and MRAP2 at the tissue level, and in individual cells in the arcuate nucleus of the hypothalamus. On the basis of the latter finding, the authors then tested whether icv injection of PK2 had differential effects on short-term food intake in wild-type and MRAP2 KO mice. PK2 appeared to be more potent in the MRAP2 KOs. Although somewhat supportive of the in vitro work performed with transfected CHO cells, clearly much more work beyond the scope of this manuscript will need to be done to further explore the interactions of PKRs and MRAP2 in vivo.

2) The authors use a MRAP2 antisera to demonstrate that neurons co-express PRK1 mRNA. The photos provided are somewhat difficult to discern double labeling. More importantly, the authors demonstrate specificity of the antisera by using transfected cells. It is unclear why knockout brain tissue was not used to validate the antisera.

3) Some more information is needed regarded the food intake paradigms. Specifically, is PK2 known to cause non-specific effects such as malaise or sickness like behavior when given at higher doses? If this has been described in the literature (or not) should be discussed at a minimum.

4) Another concern with the Introduction to the paper is why the authors focused on PKRs in the first place. There are many GPCRs to test, but no specific rationale was provided to go after PKRs in the first place. It might make more sense to first present the expression data in Figure 4 as Figure 1, then proceed with the other data figures.

5) Another concern is the immunoprecipitation studies of Figure 1. In particular, the HA-PKR1 that co-immunoprecipitates with MRAP2-Flag appears as three bands, which are labeled as "Glyco PKR1" and Non-GlycoPKR1". The immunoprecipitated HA-PKR1 of panel C also appears as multiple bands. Would those bands collapse to one band with the predicted molecular weight of PKR1 when the immunoprecipitated samples are de-glycosylated? In the absence of such experiments the designation of the bands remains to be demonstrated. Similar considerations apply to the MRAP2 bands and, in Figure 3, to the PKR2 bands. In general, given the multiple bands, the IP experiments should be validated by carrying out the Western Blot analysis with primary antibodies directly conjugated to POD, or by other approaches to eliminate the confusing contribution of the background IgG bands.

6) In Figure 2, the meaning of *** in panels G and H is unclear. Is this a post-hoc test, comparing what to what? Panels B and C are missing a control of PKR2 and mMRAP2 at ratio of 1:0. Vmax is cleared decreased in a dose response manner, what about ED50? This point is worth some discussion in the text.

7) In Figure 3 was the hMRAP2 construct used for these experiments with PKR2, but in Figure 2, mMRAP2 was used in combination with PKR2?

8) Figure 5. It would be preferable to indicate the actual amount of PK2 administered, rather than the concentration of the solution. The exact statistical comparisons used here are not clear. There are multiple doses of PK2, multiple time points, two genotypes of mice and two sexes. In addition, the comparison between saline and drug is a repeated measure for each mouse. Please expand on the statistical analyses here beyond a one-way ANOVA as indicated in the figure legend. What is the meaning of * and ** over the bars?

---

## [Author Response]

*1) The majority of the signaling and interaction data is done using heterologous cell systems. While this is clearly useful and understandable the in vivo data is less developed. They report extensive co-expression of the PKRs and MRAP2 at the tissue level, and in individual cells in the arcuate nucleus of the hypothalamus. On the basis of the latter finding, the authors then tested whether icv injection of PK2 had differential effects on short-term food intake in wild-type and MRAP2 KO mice. PK2 appeared to be more potent in the MRAP2 KOs. Although somewhat supportive of the in vitro work performed with transfected CHO cells, clearly much more work beyond the scope of this manuscript will need to be done to further explore the interactions of PKRs and MRAP2 in vivo.*

We appreciate that most of the in-vitro work is done in CHO cells and in order to confirm our findings in a more relevant system we tested the effect of MRAP2 on PKR1 trafficking and signaling in the GT1-1 hypothalamic neuronal cell line. Additionally, because another group suggested that PKR1 might be downstream of MC4R, we tested the requirement of MC4R signaling for the PKR1-mediated decrease in food intake by using MC4R KO mice. Finally, we agree with the reviewers that more work needs to be done on the role of PKR1 and MRAP2 in- vivo and that these studies are indeed out of the scope of this manuscript, however, it will be addressed in a following study on the physiology of the PKR1/MRAP2 complex.

*2) The authors use a MRAP2 antisera to demonstrate that neurons co-express PRK1 mRNA. The photos provided are somewhat difficult to discern double labeling. More importantly, the authors demonstrate specificity of the antisera by using transfected cells. It is unclear why knockout brain tissue was not used to validate the antisera.*

We agree with the reviewers that being able to use the MRAP2 KO as a negative control for the immunofluorescence would have been ideal, however, like we now clarify in the body of the manuscript, the MRAP2 KO still produces a non-functional but immunogenic fragment of MRAP2. In order to further prove the specificity of the MRAP2 antisera (in addition to the western blot previously included) we validated it for immunofluorescence using control GT1-cells (not expressing MRAP2) and GT1-1 line stably expressing MRAP2.

We also understand why the reviewers found the PKR1 staining too weak and not convincing but we were limited to in-situ hybridization due to the lack of a validated antibody raised against PKR1. In order to definitively establish the colocalization of PKR1 and MRAP2 in the brain we took advantage of a new mouse model we recently generated using CRISPR/Cas9 in which we inserted 2HA tags after the start codon of the endogenous gene, thus allowing us to detect PKR1 with the highly specific anti-HA antibody without perturbing PKR1 expression level, localization or function We hope the new microscopy results will be satisfactory.

*3) Some more information is needed regarded the food intake paradigms. Specifically, is PK2 known to cause non-specific effects such as malaise or sickness like behavior when given at higher doses? If this has been described in the literature (or not) should be discussed at a minimum.*

We now discuss the fact that in our hands and in published work, PK2 injection has not been observed to cause malaise or sickness like behavior. It was an important point, thank you for bringing it up.

*4) Another concern with the Introduction to the paper is why the authors focused on PKRs in the first place. There are many GPCRs to test, but no specific rationale was provided to go after PKRs in the first place. It might make more sense to first present the expression data in Figure 4 as Figure 1, then proceed with the other data figures.*

This is a good point and even though we mention that we did test numerous GPCRs involved in energy homeostasis and found that several were not regulated by MRAP2 (specifically GLP1R, Y2R, Y5R and MC3R) we can see how this may feel random. For this reason, we went ahead and followed your advice to move Figure 4 to Figure 1.

*5) Another concern is the immunoprecipitation studies of Figure 1. In particular, the HA-PKR1 that co-immunoprecipitates with MRAP2-Flag appears as three bands, which are labeled as "Glyco PKR1" and Non-GlycoPKR1". The immunoprecipitated HA-PKR1 of panel C also appears as multiple bands. Would those bands collapse to one band with the predicted molecular weight of PKR1 when the immunoprecipitated samples are de-glycosylated? In the absence of such experiments the designation of the bands remains to be demonstrated. Similar considerations apply to the MRAP2 bands and, in Figure 3, to the PKR2 bands. In general, given the multiple bands, the IP experiments should be validated by carrying out the Western Blot analysis with primary antibodies directly conjugated to POD, or by other approaches to eliminate the confusing contribution of the background IgG bands.*

It is true that even though most GPCRs run as several bands and that the high molecular weight smear has been shown to be a glycosylated form for many of them, we did not provide proof that it was the case for PKR1 and PKR2. In response to your comment we tried different approaches. First, we mutated the asparagine of the 2 predicted glycosylation sites to glutamine in order to prevent glycosylation of the receptors but those mutations caused the receptors to not be expressed. We then attempted to deglycosylate the receptors with PNGase F and with an Endo F1, F2, F3 mix but the receptors were mostly resistant to enzymatic deglycosylation. Because the glycosylation state of the receptors is irrelevant to our finding that MRAP2 interacts and regulate PKR1 in-vitro and in-vivo, and because we could not prove that the high molecular weight smears were glycosylated forms of the receptor, we opted to remove the label “glycosylated PKRs” from the figures. We were, however, successful in deglycosylating MRAP2 and showing that all bands collapse to the lowest molecular weight band. This result is now included in Figure 2. Finally, to eliminate the confusion brought by the IgG bands on the co-immunoprecipitation blots, we repeated the experiments but immunoprecipitated with a mouse antibody and blotted with a rabbit primary, allowing us to use an anti-rabbit secondary that does not detect the mouse antibody used for the IP.

*6) In Figure 2, the meaning of *** in panels G and H is unclear. Is this a post-hoc test, comparing what to what? Panels B and C are missing a control of PKR2 and mMRAP2 at ratio of 1:0. Vmax is cleared decreased in a dose response manner, what about ED50? This point is worth some discussion in the text.*

The statistics are now more detailed in the Methods section and the values that are compared are now linked by solid lines for clarity. We now discuss the effect of MRAP2 on PKR1 and PKR2 potency in the Results section. The “PKR2” label in the former Figure 2 was a typo and was supposed to read “PKR1” and the control for PKR1 is present in the same graph. Thank you for noticing our mistake and allowing us to fix it.

*7) In Figure 3 was the hMRAP2 construct used for these experiments with PKR2, but in Figure 2, mMRAP2 was used in combination with PKR2?*

In the former Figure 3 we used hMRAP2 with PKR2. Also, in our previous Figure 2, PKR2 was a typo – it was in fact PKR1 that was tested with both human and mouse MRAP2. Thank you for noticing this mistake and allowing us to fix it.

8) Figure 5. It would be preferable to indicate the actual amount of PK2 administered, rather than the concentration of the solution. The exact statistical comparisons used here are not clear. There are multiple doses of PK2, multiple time points, two genotypes of mice and two sexes. In addition, the comparison between saline and drug is a repeated measure for each mouse. Please expand on the statistical analyses here beyond a one-way ANOVA as indicated in the figure legend. What is the meaning of * and ** over the bars?

The actual amount of PK2 is now indicated and the statistical analysis has been clarified in the Methods. Compared values are linked with lines in the figure.